# Can the Performance Gap between Women and Men be Reduced in Ultra-Cycling?

**DOI:** 10.3390/ijerph17072521

**Published:** 2020-04-07

**Authors:** Sabrina Baumgartner, Caio Victor Sousa, Pantelis T. Nikolaidis, Beat Knechtle

**Affiliations:** 1Medbase St. Gallen Am Vadianplatz, 9001 St. Gallen, Switzerland; sabrinajosefine@hotmail.com; 2Health Technology Lab, College of Arts, Media and Design; Bouvé College of Health Sciences, Northeastern University, Boston, MA 02115, USA; cvsousa89@gmail.com; 3Exercise Physiology Laboratory, 18450 Nikaia, Greece; pademil@hotmail.com; 4Institute of Primary Care, University of Zurich, 8091 Zurich, Switzerland

**Keywords:** ultra-endurance, performance, gender, cycling speed

## Abstract

This study examined a large dataset of ultra-cycling race results to investigate the sex difference in ultra-cycling performance (100 to 500 miles) according to age and race distance. Data from the time period 1996–2018 were obtained from online available database of the ultra-cycling marathon association (UMCA), including distance-limited ultra-cycling races (100, 200, 400, and 500 miles). A total of 12,716 race results were analyzed to compare the performance between men and women by calendar year, age group (18–34, 35–44, 45–59, and 60+ years), and race distance. Men were faster than women in 100 and 200 mile races, but no sex differences were identified for the 400 and 500 mile races. The performance ratio (average cycling speed_men_/average cycling speed_women_) was smaller in the 200 mile races compared to the 100 mile races and remained stable in the 400 and 500 mile races. In all race distances, the difference in average cycling speed between women and men decreased with increasing age. The gender gap in performance was closed in several distance-limited ultra-cycling races, such as the 400 and 500 mile races.

## 1. Introduction

Ultra-endurance events (e.g., ultra-running, ultra-cycling) have gained increasing popularity over the last 25 years, with a rising number of female and master athlete participants in particular [1,2]. By definition, and according to the Ultra Marathon Cycling Association (UMCA, www.ultracycling.com), an ultra-cycling race is a race of more than 100 miles or lasting more than six hours in duration [3]. In ultra-cycling races, race distances spread from six-hour challenges up to 5000 km, such as the well-known Race Across America (RAAM). In this sports discipline, the participation of women ranges from 3% to 11%, depending on the race distance [2]. The majority of successful finishers in ultra-cycling events are master athletes in the age group of 35–49 years [1].

Sex differences in sports science have been well investigated [4,5,6,7]. The extent of the performance sex difference depends on the sport modality [8]. In ultra-cycling, women are slower than men overall [9]. Regarding an analysis of the “RAAM” (5000 km), the “Furnace Creek” 508 (800 km), and the “Swiss Cycling Marathon” (715 km), the average sex difference in ultra-cycling represents ~18%–28% [2]. An investigation of performance trends in the “Swiss Cycling Marathon” (62–560 miles) showed that sex differences decreased over the years 2001 to 2012, reaching ~14% in 2012 [10].

Sex differences in ultra-endurance performance were also investigated for other sports disciplines. Performance sex differences in running vary by race distance, where the longer the race distance, the smaller the sex difference [11]. According to iaaf.org, the gender gap in the 100 m sprint lies at ~10%, which decreases to ~9% at a marathon distance and to ~6% in ultra-marathon running (iau-ultramarathon.org). In ultra-running, the sex difference lies at ~11%–12% [12,13,14]. In long-distance triathlons, specifically the Ironman record in “Ironman Hawaii”, an even smaller sex difference of ~10.3% in performance was identified [5]. Furthermore, the gender gap was able to be closed in several time-limited ultra-marathons over the past 40 years, such as 6, 72, 144, and 24 h races [11]. Several studies in swimming demonstrated that the gender gap could be closed in ultra-endurance sports with increasing age [15,16]. In ultra-cycling, however, there is still a lack of knowledge regarding whether sex differences can be reduced with increasing age [12]. 

Aging is accompanied by a decline in physical function. This decline in performance is due to physiological mechanisms, such as a decrease in maximum aerobic capacity (VO_2max_), lactate threshold, exercise economy [17], and skeletal muscle mass [18]. Interestingly, the age-related performance decline depends on the mode of locomotion. The smallest age-related decline in performance was found in cycling when comparing swimming, running, and cycling [19]. Tanaka and Seals [17] demonstrated a curvilinear decrease in age-related performance up to the age of 60 years, whereupon performance decreased exponentially thereafter when analyzing results in running and swimming for both genders. Ransdell et al. [8] analyzed cycling results (i.e., 200 m and 500 m track and a 40 km road time-trial) and showed that this exponential decline after the age of 55 years occurred to a greater extent in female athletes. Spina et al. [20] found that both the cardiovascular response and adaption to endurance exercise training are better in elderly men than in elderly women. 

The age at peak performance and the sex differences in ultra-endurance performance vary between sports. In ultra-cycling, the age at peak performance seemed to be similar for both sexes, specifically ~37 for women and ~38 for men, according to an analysis of a 24-hour ultra-cycling draft-legal event held in Switzerland [10]. In 50 km ultra-running races, men achieved peak performance at a younger age (30–39 years) than women (40–49 years) [21]. This was in contrast to ultra-swimming, where no sex difference in peak performance was demonstrated in the elderly age groups (30–39 years, 40–49 years, and 50–59 years) [22,23]. 

Based on existing literature, we know that female endurance performance generally declines to a greater extent compared to male endurance performance [8]. This work relied on an enormous number of ultra-cycling races to analyze trends in ultra-cycling with special regard to the gender gap according to race distance and age. This is the first study to include such a large number of participants throughout different race distances (100–500 miles) [10]. 

## 2. Materials and Methods 

### 2.1. Ethical Approval

All procedures used in this study were approved by the Institutional Review Board of Kanton St. Gallen, Switzerland, with a waiver of the requirement for informed consent of participants given the fact that the study involved the analysis of publicly available data. 

### 2.2. Data Sampling 

Data were collected from publicly available data (World Ultra Cycling Association) [24]. We used the calendar of the Ultramarathon Cycling Association to include the most common races in ultra-cycling all over the world. Data from 1996 to 2018 were obtained from the official race websites. Required information in the original dataset included year of competition, gender, age group, race distance, and racing time. Other data were excluded due to vague age grouping. Based on the different race distances, cycling speed was used as a comparable variable. Data of 12,716 athletes throughout 45 different events was analyzed (Table A1). The whole dataset was cleaned for double coding results. Unfortunately, much of the European data was removed due to missing necessary information. For example, known races such as the Race across Italy, Tortour in Switzerland, the Race across Germany, and Slo24 Ultra had to be excluded. We did not distinguish between nationalities in this study. 

### 2.3. Statistical Analysis

Data were tested for normality and homogeneity using the Shapiro–Wilk and Levene’s tests, respectively. Data were expressed as means and standard deviations (±SD). An analysis of variance (one-way ANOVA) compared the performance between different age groups and different race distances. Additionally, Student’s t-test for independent samples was applied to compare performances between men and women. Three general linear models were performed (two-way ANOVA) according the following interactions: model 1 = gender × year; model 2 = gender × race distance; model 3 = gender × age group. Performance ratios were calculated according to male and female performance (average cycling speed_men_/average cycling speed_women_), with a higher performance ratio indicating a greater sec difference. Statistical significance was set at 5% (*p* < 0.05). The Statistical Software for the Social Sciences (SPSS v23.0 IBM, Chicago, IL, USA) and GraphPad Prism v.6.0 (GraphPad Software, San Diego, CA, USA) were used for all analyses.

## 3. Results

The number of participating athletes by calendar year for men and women showed greater male participation in all distance races, with 11,347 (91.05%) men and 1116 (8.95%) women (Figure 1).

In the first model (gender × year), a gender-effect was identified in 200 (F = 22.0, *p* < 0.001 and 400 mile (F = 7.4, *p* = 0.010) races. Moreover, a year effect and interaction between gender and year were found for 200 (F = 9.2, *p* < 0.001; F = 2.3, *p* < 0.001) and 400 (F = 3.3, *p* = 0.014; F = 1.9, *p* = 0.025) mile races, respectively (Figure 2). 

The second model (gender and race distance) showed a trend for a race distance effect (F = 6.0, *p* = 0.087) and an interaction effect (F = 15.4, *p* < 0.001); pairwise comparisons showed that men were faster than women in 100 mile and 200 mile races (Figure 3). The third model (gender and age group) showed a significant gender effect for 100 mile races (F = 9.1, *p* = 0.026), and age group effects were identified in 200 mile races (F = 9.1, *p* = 0.028). Interaction effects were shown in 100 mile (F = 7.1, *p* < 0.001) and 200 mile (F = 4.1, *p* = 0.002) races. Post-hoc analyses showed a trend regarding men in the younger age group being faster (*p* < 0.05) than the adjacent older age group in 100 mile and 200 mile races, but no differences were observed between the 18–34 and 35–44 age groups in 400 mile and 500 mile races. For the women, the fastest age group in the 100 miles races was 45–59 and 35–44 in the 400 mile races. For the 200 miles races, the female younger age group was faster (*p* < 0.05) than the adjacent older age group (Figure 4). Moreover, post-hoc analyses showed that men were faster in the 18–34, 35–44, and 60+ age groups in 100 miles races; in 200 mile races, men were faster only in the 45–59 age group, and in 400 and 500 mile races men were faster only in the 18–34 age group.

## 4. Discussion

We investigated the impact of age, gender, and race distance on performance in ultra-cycling by analyzing a large dataset including 12,716 results from 45 different ultra-cycling races from 1996 to 2018. We hypothesized a decrease in the gender gap with increasing age and increasing race distance. The main findings were that (i) men had the highest participation in ultra-cycling races over all distances, (ii) men were faster than women over all distances in the majority of the years, (iii) men were faster than women in 100 and 200 mile races, (iv) no sex differences were identified for the 400 and 500 mile races, (v) the performance ratio decreased from the 100 mile races to the 200 mile races and remained stable in the 400 and 500 mile races, and (vi) men showed a higher average cycling speed than women over all race distances (100–500 mile races), with the difference in average cycling speed between women and men decreasing with increasing age. To sum up, the gender gap in ultra-cycling narrowed with increasing age and increasing racing distance. 

### 4.1. The Gender Gap was Reduced in Longer Ultra-Cycling Race Distances

An important finding was that the gender gap was able to be reduced in longer distance-limited ultra-cycling races (200, 400, and 500 mile races). Previous studies struggled to examine whether women could outperform men in the future [4]. Physiological, hormonal, and genetic factors responsible for gender differences must be considered [25,26], such as women having smaller hearts, lower bone density, shorter stride length, lower cardiac output, smaller lungs, lower lung capacity, smaller muscles in relation to body size [19], lower testosterone levels resulting in higher body fat percentage, and lower hemoglobin concentrations resulting in lower anaerobic capacity [25,27,28].

However, in ultra-endurance sports, some of the above-mentioned anthropometric and physiological characteristics can be of advantage, for example, a higher percentage of body fat serving as an energy store [29]. Furthermore, ultra-endurance sports are less dependent on anaerobic energy supply, which is preserved by the impact of testosterone. Physiological conditions result in a gender gap that is the highest over short distances and drops with increasing running distance (according to iaaf.org and iau-ultramarathon.org). Similar findings in this study were that men were faster than women in shorter ultra-cycling distances, such as 100 and 200 mile races, but no significant differences were identified in the 400 and 500 mile races. Thus, gender has an influence on performance, with the performance ratio (PR, average cycling speed_men_/average cycling speed_women_) dropping from 100 mile (PR: 1.1633) to 200 mile races (PR: 1.0285) and remaining stable in 400 (PR: 1.0217) and 500 mile (PR: 1.0403) races; therefore the gender gap was greatest in the 100 mile races. 

### 4.2. Participation Patterns in Ultra-Cycling According to Gender 

The number of participating athletes in distance-limited ultra-cycling races was unstable, with the highest participation of men over all race distances; there were 11,347 (91.04%) men and only 1116 (8.95%) women (Figure 1). This finding was congruent with other investigations regarding participation trends in ultra-cycling events [1,2] where ~3–11% were female finishers. This lag in participation could partially be explained by the fact that women were not allowed to participate in sports competitions for many years due to previous emancipation, historical, and social reasons [2]. Furthermore, the gap in participation could be explained by different exercise behaviors between women and men, whereby men seem to seek more strenuous exercise involvement [30].

### 4.3. Average Cycling Speed and Performance Ratio According to Gender and Age

The average cycling speed dropped from one age group to another in 100 and 200 mile races, but there was no similar trend in the 400 and 500 mile races. A similar conclusion was drawn in a former study investigating ultra-cycling races, indicating that the fastest cyclists were younger and participated in shorter races [31]. The performance ratio was the lowest for the age category 34–44 years (PR: 0.8163), increasing for the age group 45–59 years (PR: 0.8818), and even more so for the 60+ age group (PR: 0.9178), thereby showing the narrowing of the gender gap in ultra-cycling according to increasing age. 

### 4.4. Limitations, Strengths, and Implications for Future Research

The strength of this study lies in the large dataset including different race distances, different races, and a huge number of athletes of different performance levels studied over a longer period of time (1996–2018). The difference and the ratio between men and women should be interpreted with caution since there were significantly more data (participants) for men, which may have inappropriately skewed performance data in the small sub-analyses. This cross-sectional study was limited because we were unable to take into consideration other aspects influencing cycling performance, such as previous training, equipment, weight, weather, topographic characteristics, nutrition, and motivation. Nevertheless, this additional information would not have changed our results because we compared the performance gap between men and women. In this study, we demonstrated the influence of age on the gender gap. The fact that both the performance ratio and the gender gap were smallest in the longest race distances and the oldest athletes may empower athletes older than 45 years to compete in longer race distances. 

## 5. Conclusions

In distance-limited ultra-cycling races (100–500 miles) held from 1996 to 2018, men were faster than women for all race distances. Men showed greater participation than women. The gender gap was the highest in the 100 mile races and narrowed in longer race distances (200, 400, and 500 miles). The gender gap decreased in distance-limited ultra-cycling races when race distance and age increased. This finding may empower women to compete in longer race distances to keep up with male competitors. Further studies are necessary to investigate the high participation lag in ultra-cycling races between men and women. Furthermore, it would be of interest to know whether the gender gap equally narrows in time-limited ultra-cycling races and in ultra-cycling races longer than 500 miles, such as the RAAM. 

## Figures and Tables

**Figure 1 ijerph-17-02521-f001:**
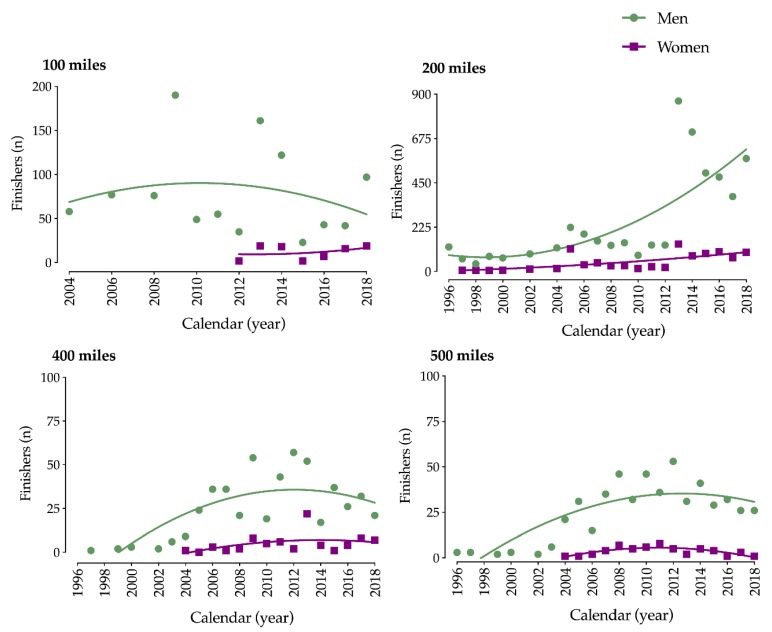
Number of male and female participants in ultra-distance cycling in 100–500 mile races from 1996 to 2018. * Outlier point (n = 190).

**Figure 2 ijerph-17-02521-f002:**
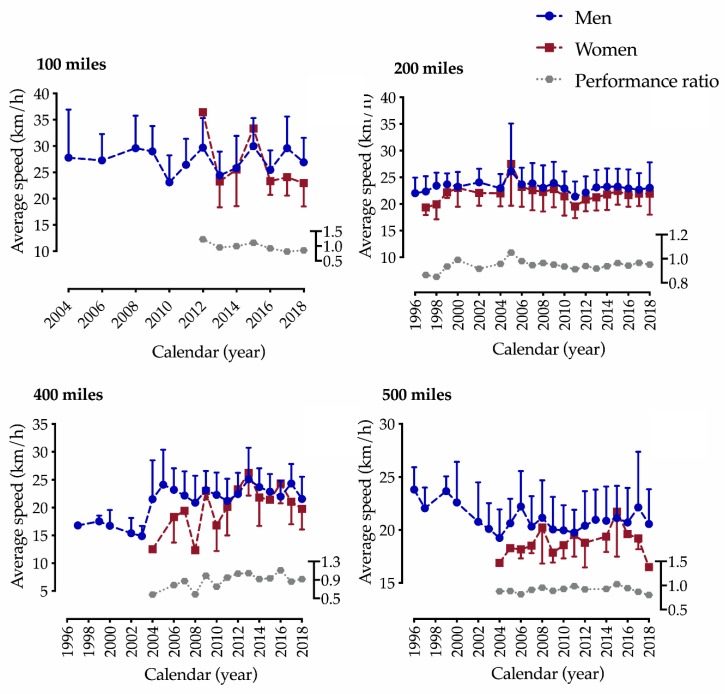
Performance trends of male and female participants in ultra-distance cycling in 100–500 mile races from 1996 to 2018. Data expressed as means and standard deviations.

**Figure 3 ijerph-17-02521-f003:**
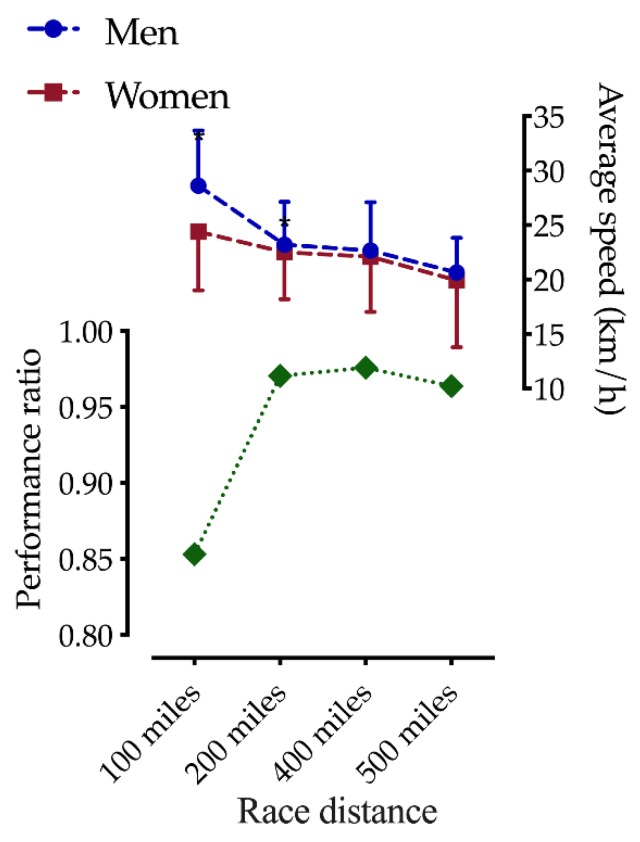
Comparison of performance level between men and women participating in ultra-distance cycling (100–500 mile races). * Statistical difference between men and women. Data expressed as means and standard deviations.

**Figure 4 ijerph-17-02521-f004:**
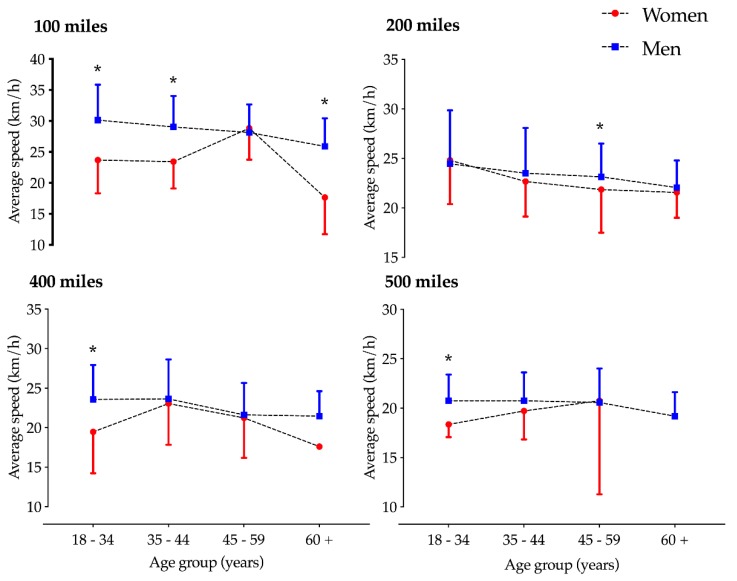
Performance trend by gender and age groups of participants in ultra-distance cycling (100–500 mile races). * Significant difference between men and women (*p* < 0.05) in post-hoc analysis.

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
