# Peer review of "Can the Performance Gap between Women and Men be Reduced in Ultra-Cycling?"

_ijerph, 2020, doi:10.3390/ijerph17072521_

Round 1
Reviewer 1 Report
Dear Authors,
Thank you for a largely well prepared paper that has examined a very large dataset and made efforts to distinguish between performances across age groups and between genders. The analysis is commendable and the data are also presented in a way that makes the paper easier to understand.
My main criticism of the paper however, is that throughout it reads as if a draft or notes of a paper in progress - this can largely be addressed by combining sentences that discuss similar points, but the style/pattern is consistent throughout. You also state performing an ANOVA, but it appears that this is only correctly reported in the figures - I recommend including p values, and performance ratio in parentheses after results statements. An estimate of effect size is also recommended, as this prevents the presentation of a false dichotomy e.g. males were/were not quicker than females.
Line by line recommendations:
Line 28 change women to female
Lines 29 - 32 consider combining these sentences, given their common definition
Line 36/37 is this statement regarding elderly athletes relevant to the current discussion/scope of the paper, especially as it is not supported by a reference?
Line 38: Consider rewording to 'Sex differences in sport science have been well investigated [4-7]'
Line 40-42 please reorder the sentence regarding RAAM analyses, as it currently isn't quite clear what this means
Line 44 Sex differences have decreased over the years 2001-2012....
The following paragraphs (Lines 46-78) involve a lot of repetition, I'd consider how you demonstrate that the trends are similar across exercise modalities etc. and how these can be best combined without sounding too repetitive, it may be as simple as focusing on a small number of key references, but citing multiple references that agree with your position as opposed to the current approach of giving equal space to each reference
Line 80-81 why is the need to have knowledge about the performance trends for masters cyclists important? Can you please add some context here e.g. are participation numbers increasing? Do they have greater disposable income? Is it a vehicle for healthy ageing?
Line 102 'because of vague age group' --> because of vague age grouping
Line 102/103 consider placing this sentence earlier e.g. Line 98?
Line 115/116 please rework this sentence for clarity
Results generally - please state p values and performance ratios when appropriate, as opposed to simply referencing a figure. Greater precision needed throughout results in this regard. It may mean doubling the word count of your results, but they are likely to be better understood
Line 146 - amend to 'results from 46 different...'
Line 154 - can you be specific here, Figure 3 and 5 seems to suggest that this depends upon distance and age of participants, let the reader know exactly where differences occur and to what extent
Line 159-163 - are all of these physiological differences between sexes from reference [22]? If so, it may need referencing again at the end of this paragraph for completeness
Line 171-175 - these sentences can be linked to provide more flow to the conclusion of this paragraph
Line 185 - please provide more context around reference [24]. It's insufficient in a paper that discusses gender AND age differences to simply state men prefer harder exercise. Consider when each gender likely participates or drops out of and returns to exercise, and the nature of exercise each gender is attracted to, not simply reporting one side of the story
Lines 204-207 this statement regarding an individual athlete here either needs greater context (perhaps placing it earlier in the discussion) or should not be included
Please also include reference to figures where appropriate throughout the discussion and check website referencing format as per MDPI guidelines
Author Response
Reviewer 1
Dear Authors,
Thank you for a largely well-prepared paper that has examined a very large dataset and made efforts to distinguish between performances across age groups and between genders. The analysis is commendable and the data are also presented in a way that makes the paper easier to understand.
My main criticism of the paper however, is that throughout it reads as if a draft or notes of a paper in progress - this can largely be addressed by combining sentences that discuss similar points, but the style/pattern is consistent throughout. You also state performing an ANOVA, but it appears that this is only correctly reported in the figures - I recommend including p values, and performance ratio in parentheses after results statements. An estimate of effect size is also recommended, as this prevents the presentation of a false dichotomy e.g. males were/were not quicker than females.
Answer: We agree with the expert reviewer and we have inserted p-values as recommended.
Line 28 change women to female
Answer: We agree with the expert reviewer and changed it to female.
Lines 29 - 32 consider combining these sentences, given their common definition
Answer: We agree with the expert reviewer and combined these sentences.
Line 36/37 is this statement regarding elderly athletes relevant to the current discussion/scope of the paper, especially as it is not supported by a reference?
Answer: We agree with the expert reviewer and took this sentence out.
Line 38: Consider rewording to 'Sex differences in sport science have been well investigated [4-7]'
Answer: We agree with the expert reviewer and have changed it as recommended.
Line 40-42 please reorder the sentence regarding RAAM analyses, as it currently isn't quite clear what this means
Answer: We agree with the expert reviewer and have reordered the sentence for better understanding.
Line 44 Sex differences have decreased over the years 2001-2012....
Answer: We agree with the expert reviewer and have changed it as recommended.
The following paragraphs (Lines 46-78) involve a lot of repetition, I'd consider how you demonstrate that the trends are similar across exercise modalities etc. and how these can be best combined without sounding too repetitive, it may be as simple as focusing on a small number of key references, but citing multiple references that agree with your position as opposed to the current approach of giving equal space to each reference
Answer: We agree with the expert reviewer and have shortened resp. trimmed this passage for better understanding.
Line 80-81 why is the need to have knowledge about the performance trends for masters cyclists important? Can you please add some context here e.g. are participation numbers increasing? Do they have greater disposable income? Is it a vehicle for healthy ageing?
Answer: We agree with the expert reviewer and took this passage out. It is not fitting the context.
Line 102 'because of vague age group' --> because of vague age grouping
Answer: We agree with the expert reviewer and changed it as recommended.
Line 102/103 consider placing this sentence earlier e.g. Line 98?
Answer: We agree with the expert reviewer and placed this sentence as recommended.
Line 115/116 please rework this sentence for clarity
Answer: We agree with the expert reviewer and changed it. We have included further information (amount of male and female participants in percent).
Results generally - please state p values and performance ratios when appropriate, as opposed to simply referencing a figure. Greater precision needed throughout results in this regard. It may mean doubling the word count of your results, but they are likely to be better understood
Answer: We agree with the expert reviewer and p-values have been inserted accordingly.
Line 146 - amend to 'results from 46 different...'
Answer: We agree with the expert reviewer and have changed it as recommended.
Line 154 - can you be specific here, Figure 3 and 5 seems to suggest that this depends upon distance and age of participants, let the reader know exactly where differences occur and to what extent
Answer: We agree with the expert reviewer and changed the last two findings appropriately. To complete we sum up the most important findings in the last sentence for better understanding.
Line 159-163 - are all of these physiological differences between sexes from reference [22]? If so, it may need referencing again at the end of this paragraph for completeness
Answer: We agree with the expert reviewer and added further references for completeness.
Line 171-175 - these sentences can be linked to provide more flow to the conclusion of this paragraph
Answer: We agree with the expert reviewer and have changed it as recommended.
Line 185 - please provide more context around reference [24]. It's insufficient in a paper that discusses gender AND age differences to simply state men prefer harder exercise. Consider when each gender likely participates or drops out of and returns to exercise, and the nature of exercise each gender is attracted to, not simply reporting one side of the story
Answer: We agree with the expert reviewer in general, but we didn`t find further explanations in literature for this participation lag. It was written “gender gap” but meant “participation gap” and we changed it to this.
Lines 204-207 this statement regarding an individual athlete here either needs greater context (perhaps placing it earlier in the discussion) or should not be included
Answer: We agree with the expert reviewer and removed this text passage.
Please also include reference to figures where appropriate throughout the discussion and check website referencing format as per MDPI guidelines
Answer: We agree with the expert reviewer and changed as requested
Reviewer 2 Report
Please reexamine the description in the introduction, methods, results, discussion and conclusions.
This article has the following many serious problems and was judged to be at inappropriate level as an original paper.
1. Introduction
The gender and performance differences in relation to race distance of ultra-cycling, which are the purpose of this study, may have already clarified in Reference 2, 9 and 10.
2.2. Data sampling
・As the basic information, the number and average age of male and female subjects in each age group and race distance should be described.
・It is written that data from 1983 to 2018 was used, but Fig.1 shows only data from 1996 to 2018.
・When only participants were male, it was unclear that how was data processed.
2.3. Statistical analysis
・The authors should test the gender differences in different age groups and race distances by a two-way analysis of variance?
・It should be noted that the performance ratio was calculated as average race speed in men/average race speed in women.
3. Results
・Why are the age intervals different from each other?. The age intervals should have been divided into 18-29, 30-39, 40-49, 50-59, and over 60 years old from standpoint of comparison with previous research results.
・In some years, women run faster than men (Figure 2). The description that " Men were faster than women in all distance throughout the years" may be incorrect.
・The authors should test main effects and interactions in changes of average race speed of men and women with increase of race distance(Figure.3).
・In Figure 4, age-related change of average speed should be shown for men and women, with increase of age should be shown
・In Figure 5, the comparison of average cycling speeds between different age groups should be performed from standpoints of race distance and gender. In addition, the change of performance ratio in lower panel is different from above-mentioned definition.
・The description "Men showed a higher average cycling speed than women in race distances (100- to 500-mile race), with a decreasing performance ratio in the 200-mile (Figure 5)" can not be derived from Figure 5.
・Average speeds in Fig. 4 and Fig.5 should be shown by gender and race distance separately, and analyzed for main effects and interractions.
4. Discussion
4.1. Women reduce the gender gap in longer ultra-cycling in race distance
・There is no discussion on the result that the gender differences in performance observed in this study were smaller than those of previous studies. Therefore, the validity of results does not remain unclear.
4.3. Average cycling speed and performance ration in dependence of gender and age
・The authors should clarify how authors handled male data in the calendar year when there was no female participation.
・According to the definition of performance ratio(men/women), the description " Comparing gender, performance ratio dropped down for the age category 34-44 years, followed by a significant increase from the age of 45 years" is not correct.
・It should be discussed whether the performance ratio between men and women have changed significantly only at the age of 34-45 years.
4.4. Limitations, strength and implications for future research
・The authors should show that the limitations presented do not significantly influence the conclusion in your study.
5. Conclusions
・It may be inappropriate statement that "In distance limited ultra-cycling races held from 1983 to 2018", because the data from 1983 to 1995 was not showed.
・The hypothesis of this study was followed as the gender gap will narrow the longer the race distance. However, there is almost no change in the performance ratio from 200 to 500 miles. Therefore, the conclusion that "Female ultra-cyclist seem to be able to narrow the gap to men in longer ultra-marathon cycling distance" should be reexamined
Author Response
Reviewer 2
Please reexamine the description in the introduction, methods, results, discussion and conclusions.
Answer: We agree with the expert reviewer and adapted those.
This article has the following many serious problems and was judged to be at inappropriate level as an original paper.
Answer: We agree with the expert reviewer and improved the paper as recommended.
1. Introduction
The gender and performance differences in relation to race distance of ultra-cycling, which are the purpose of this study, may have already clarified in Reference 2, 9 and 10.
Answer: The purpose of this study is to examine gender gap mainly in dependence of increasing age and racing distance in ultra-cycling. There exist papers about gender difference in general (Reference 2, 9 and 10) in ultra-cycling but there doesn`t exist a paper that answers the dynamic of gender difference in dependence of age and racing distance.
2.2. Data sampling
・As the basic information, the number and average age of male and female subjects in each age group and race distance should be described.
Answer: We agree with the expert reviewer and amended the number of male and female participants. Unluckily it is not possible to mention average age because most of the races make just age-group available.
・It is written that data from 1983 to 2018 was used, but Fig.1 shows only data from 1996 to 2018.
Answer: We agree with the expert reviewer and changed it, the data of the year 1983 had to be excluded because of missing age of the participants.
・When only participants were male, it was unclear that how was data processed.
Answer: Thank you for your input, we added this information in “Statistical Analysis” (last sentence): In year without female data, men data was handled in separate models (one level-no gender interaction).
2.3. Statistical analysis
・The authors should test the gender differences in different age groups and race distances by a two-way analysis of variance?
Answer: The second two-way ANOVA model (gender x race distance) show significant effect for gender (F= 36.1; p< 0.001), race distance (F= 98.1; p<0.001), and interaction (F=13.7; p< 0.001).
・It should be noted that the performance ratio was calculated as average race speed in men/average race speed in women.
Answer: We agree with the expert reviewer and added this information for better understanding.
- Results
・Why are the age intervals different from each other? The age intervals should have been divided into 18-29, 30-39, 40-49, 50-59, and over 60 years old from standpoint of comparison with previous research results.
Answer: The recommended age group is not consistent with the age-groups of the ultra-cycling association (https://ultracycling.com/)
・In some years, women run faster than men (Figure 2). The description that " Men were faster than women in all distance throughout the years" may be incorrect.
Answer: We agree with the expert reviewer and we changed this formulation to “in the majority”.
・The authors should test main effects and interactions in changes of average race speed of men and women with increase of race distance (Figure.3).
Answer: We agree with the expert reviewer.
・In Figure 4, age-related change of average speed should be shown for men and women, with increase of age should be shown
Answer: Has been adapted.
・In Figure 5, the comparison of average cycling speeds between different age groups should be performed from standpoints of race distance and gender. In addition, the change of performance ratio in lower panel is different from above-mentioned definition.
Answer: Figure 5 has been excluded.
・The description "Men showed a higher average cycling speed than women in race distances (100- to 500-mile race), with a decreasing performance ratio in the 200-mile (Figure 5)" cannot be derived from Figure 5.
Answer: We agree with the expert reviewer it cannot be derived from Figure 5 but from Figure 2. Therefore, this formulation has been changed to: Pairwise comparisons showed that men have a higher average cycling speed than women in all age-groups, and that men reach a higher speed at 35-44 before dropping, whereas women reach a peak at 18-34 (Figure 5).
・Average speeds in Fig. 4 and Fig.5 should be shown by gender and race distance separately, and analyzed for main effects and interactions.
Answer: We agree with the expert reviewer and changed as requested.
- Discussion
4.1. Women reduce the gender gap in longer ultra-cycling in race distance
・There is no discussion on the result that the gender differences in performance observed in this study were smaller than those of previous studies. Therefore, the validity of results does not remain unclear.
Answer: We agree with the expert reviewer and changed as requested.
4.3. Average cycling speed and performance ration in dependence of gender and age
・The authors should clarify how authors handled male data in the calendar year when there was no female participation.
Answer: Thank you for this input, we added this information in “Statistical Analysis”.
・According to the definition of performance ratio(men/women), the description " Comparing gender, performance ratio dropped down for the age category 34-44 years, followed by a significant increase from the age of 45 years" is not correct.
Answer: Performance ratio is only possible to compare with other age-groups. It is the ratio between averagespeedmen/averagespeedwomen. Performance ratio is a single data for each age-group.
・It should be discussed whether the performance ratio between men and women have changed significantly only at the age of 34-45 years.
Answer: We have been interested how the performance ratio changes in dependence of the racing distance. Performance ratio is an additional analysis to help to interpret the results from ANOVA. Performance ratios are absolute numbers from the data they don`t have dispersion measure. This means it does not have a statistical test to compare.
4.4. Limitations, strength and implications for future research
・The authors should show that the limitations presented do not significantly influence the conclusion in your study.
Answer: We agree with the expert reviewer and adapted this.
- Conclusions
・It may be inappropriate statement that "In distance limited ultra-cycling races held from 1983 to 2018", because the data from 1983 to 1995 was not showed.
Answer: We agree with the expert reviewer and corrected it to 1996-2018. The other data (1983-1995) had to be excluded.
The hypothesis of this study was followed as the gender gap will narrow the longer the race distance. However, there is almost no change in the performance ratio from 200 to 500 miles. Therefore, the conclusion that "Female ultra-cyclist seem to be able to narrow the gap to men in longer ultra-marathon cycling distance" should be reexamined
Answer: We agree with the expert reviewer and changed this sentence.
Reviewer 3 Report
- Page 2, line 50. “For example, according to iaaf.org, the gender gap in 100 m sprint lies at 10%, decreases in the marathon distance to 9% and to 6 % in ultra-marathon running (iau-ultramarathon.org)”
This is not a good study to refer to if the authors want to suggest a difference in sport performance. I could suggest Ospina Betancurt, et al 2018. Sex-differences in elite-performance track and field competition from 1983 to 2015. Journal of Sport Sciences 36 (11) 1262-1268.
- Page 2, line 60. “In ultra-cycling, however, there is still a lack of knowledge whether women are able to reduce the sex difference to men with increasing age”.
Regarding this phrase I would like to suggest the following references:
- Rust, CA (Ruest, Christoph Alexander)[ 1 ] ; Rosemann, T (Rosemann, Thomas)[ 1 ] ; Lepers, R (Lepers, Romuald)[ 2 ] ; Knechtle, B (Knechtle, Beat)[ 3 ]. Gender difference in cycling speed and age of winning performers in ultra-cycling - the 508-mile "Furnace Creek" from 1983 to 2012. JOURNAL OF SPORTS SCIENCES. Volumen: 33 Número: 2 Páginas: 198-210. DOI: 10.1080/02640414.2014.934705
- Pozzi, Lara; Knechtle, Beat; Knechtle, Patrizia; Rosemann, Thomas; Lepers, Romuald; Rust, Christoph Alexander. Sex and age-related differences in performance in a 24-hour ultra-cycling draft-legal event - a cross-sectional data analysis. BMC sports science, medicine & rehabilitation. Volumen:6 Páginas:19DOI:10.1186/2052-1847-6-19
- Salihu, L (Salihu, Lejla); Rust, CA (Rust, Christoph Alexander); Rosemann, T (Rosemann, Thomas); Knechtle, B (Knechtle, Beat). Sex Difference in Draft-Legal Ultra-Distance Events - A Comparison between Ultra-Swimming and Ultra-Cycling. CHINESE JOURNAL OF PHYSIOLOGY. Volumen: 59 Número: 2 Páginas: 87-99 DOI: 10.4077/CJP.2016.BAE373
- Page 2, line 81. There is an erratum “Therefore, the aim of his study…” I think is “Therefore, the aim of this study…”
- Section 2.2 Data sampling. The authors wrote “data of 12.716 athletes in 47 different events have been analyzed”. However, it would be very interesting to list all events where the data came from, for example as a table. Without this information the research cannot be replicated by other researchers.
- Figure 1. I suggest to locate the title of each figure further away from the “Y” axis. For example, the mark of "200 finishers" lies right next to the title " 100 miles" which, at first sight, is not easy to distinguish. The authors should do the same in all figures.
- Page 6, line 146. Have the authors analyzed 46 or 47 events (according to what they wrote in the data sampling section)?
- Reference 6. The title is in capital letters. Should it be in capital letters?
- Reference 9. There is an erratum. Rust, C.A instead of Ruest, C.A.
Author Response
Reviewer 3
Page 2, line 50. “For example, according to iaaf.org, the gender gap in 100 m sprint lies at 10%, decreases in the marathon distance to 9% and to 6 % in ultra-marathon running (iau-ultramarathon.org)”
This is not a good study to refer to if the authors want to suggest a difference in sport performance. I could suggest Ospina Betancurt, et al 2018. Sex-differences in elite-performance track and field competition from 1983 to 2015. Journal of Sport Sciences 36 (11) 1262-1268.
Page 2, line 60. “In ultra-cycling, however, there is still a lack of knowledge whether women are able to reduce the sex difference to men with increasing age”.
Regarding this phrase I would like to suggest the following references:
- Rust, CA (Ruest, Christoph Alexander)[ 1 ] ; Rosemann, T (Rosemann, Thomas)[ 1 ] ; Lepers, R (Lepers, Romuald)[ 2 ] ; Knechtle, B (Knechtle, Beat)[ 3 ]. Gender difference in cycling speed and age of winning performers in ultra-cycling - the 508-mile "Furnace Creek" from 1983 to 2012. JOURNAL OF SPORTS SCIENCES. Volumen: 33 Número: 2 Páginas: 198-210. DOI: 10.1080/02640414.2014.934705
- Pozzi, Lara; Knechtle, Beat; Knechtle, Patrizia; Rosemann, Thomas; Lepers, Romuald; Rust, Christoph Alexander. Sex and age-related differences in performance in a 24-hour ultra-cycling draft-legal event - a cross-sectional data analysis. BMC sports science, medicine & rehabilitation. Volumen:6 Páginas:19DOI:10.1186/2052-1847-6-19
- Salihu, L (Salihu, Lejla); Rust, CA (Rust, Christoph Alexander); Rosemann, T (Rosemann, Thomas); Knechtle, B (Knechtle, Beat). Sex Difference in Draft-Legal Ultra-Distance Events - A Comparison between Ultra-Swimming and Ultra-Cycling. CHINESE JOURNAL OF PHYSIOLOGY. Volumen: 59 Número: 2 Páginas: 87-99 DOI: 10.4077/CJP.2016.BAE373
Answer: We agree with the expert reviewer and
Page 2, line 81. There is an erratum “Therefore, the aim of his study…” I think is “Therefore, the aim of this study…”
Answer: We agree with the expert reviewer this text passage has been deleted.
Section 2.2 Data sampling. The authors wrote “data of 12.716 athletes in 47 different events have been analyzed”. However, it would be very interesting to list all events where the data came from, for example as a table. Without this information the research cannot be replicated by other researchers.
Answer: Thank you for this input. We corrected it to 45 race events. Furthermore, we added a table with all races included for better understanding.
Figure 1. I suggest to locate the title of each figure further away from the “Y” axis. For example, the mark of "200 finishers" lies right next to the title " 100 miles" which, at first sight, is not easy to distinguish. The authors should do the same in all figures.
Answer: We agree with the expert reviewer and revised as recommended.
Page 6, line 146. Have the authors analyzed 46 or 47 events (according to what they wrote in the data sampling section)?
Answer: We agree with the expert reviewer and have corrected it to 45 race events.
Reference 6. The title is in capital letters. Should it be in capital letters?
Answer: No, the title should not be in capital letters, we changed it.
Reference 9. There is an erratum. Rust, C.A instead of Ruest, C.A.
Answer: Thank you, he is called Rüest, C.A.
Round 2
Reviewer 1 Report
I recommend that the authors do a final proof-read of the text and assess their writing for English language errors, some of these appear minor but affect the 'readability' of the paper. It is however, much improved and is clear that authors have aimed to address comments from all three reviewers thoroughly.
In Figures 2 and 4, I would recommend that the speeds are converted to km/h as per SI units which MDPI journals typically abide by, as this potentially increases the transferabiltiy and impact of your findings beyond this paper - I appreciate this is an extra step in data presentation at a late stage, but I think the paper will benefit from it and continue to do so beyond publication
line 179 - 180 - the sentence beginning 'Being...' is really short and does not make sense, please tie it to former sentence in a way that complements the message you're trying to convey.
Further, this difference in weighting/percentage of each gender to the total set is an issue that needs to be discussed in the limitations as it potentially inappropriately skews the performance ratio and other statistics - especially in cases where females are faster, and where male SD's are large.
Following these changes, and a final proof read I am happy to recommend acceptance
Author Response
Reviewer 1
I recommend that the authors do a final proof-read of the text and assess their writing for English language errors, some of these appear minor but affect the 'readability' of the paper. It is however, much improved and is clear that authors have aimed to address comments from all three reviewers thoroughly.
Answer: We agree with the expert reviewer and did therefore a final proof-read.
In Figures 2 and 4, I would recommend that the speeds are converted to km/h as per SI units which MDPI journals typically abide by, as this potentially increases the transferability and impact of your findings beyond this paper - I appreciate this is an extra step in data presentation at a late stage, but I think the paper will benefit from it and continue to do so beyond publication
Answer: We agree with the expert reviewer, all data have been converted to km/h, analysis had to be ran again and graphics and results were updated.
line 179 - 180 - the sentence beginning 'Being...' is really short and does not make sense, please tie it to former sentence in a way that complements the message you're trying to convey.
Answer: We agree with the expert reviewer and connected these sentences. The number of participating athletes in distance-limited ultra-cycling races is unstable, with the highest participation of men in all race distances; being 11,347 (91.04 %) for men, and 1,116 (8.95 %) for women (Figure 1).
Further, this difference in weighting/percentage of each gender to the total set is an issue that needs to be discussed in the limitations as it potentially inappropriately skews the performance ratio and other statistics - especially in cases where females are faster, and where male SD's are large.
Answer: We agree with the expert reviewer and added this as a limitation at the end of our discussion. Please see all changes marked as red.
Following these changes, and a final proof read I am happy to recommend acceptance
Reviewer 2 Report
Reviewer 2
Please reexamine the description in the introduction, methods, results, discussion and conclusions.
Answer: We agree with the expert reviewer and adapted those.
Abstract: This study examined a large dataset of ultra-cycling race results to investigate the sex difference in ultra-cycling performance (100 to 500 miles) in dependence of age and race distance. Data from the time period 1996-2018 were obtained from the online available database of the ultra-cycling marathon association (UMCA) including distance-limited ultra-cycling races (100-, 200-, 400- and 500-miles). A total of 12,716 race results were analyzed to compare performance between men and women by calendar year, age group (18-34, 35-44, 45-59 and 60+ years), and race distance. Men were faster than women in 100- and 200-mile races. No sex differences were identified for the 400- and 500-mile races. The performance ratio (average speedmen/ average speedwomen) dropped from 100- to 200-mile race and remained stable in 400- and 500-mile. In all race distances, the difference in average cycling speed between women and men decreased with increasing age. Women were able to close the gender gap to men in several distance-limited ultra-cycling races such as 400- and 500-mile races.
Reviewer
・You need to write the conclusions of your study that answer your research purposes. The conclusions are differrent from the results.
・such as 400- and 500-mile races → Why do you exclude 200-mile race?
・The description 「In all race distances, the difference in average cycling speed between women and men decreased with increasing age.」should be reconsidered.
This article has the following many serious problems and was judged to be at inappropriate level as an original paper.
Answer: We agree with the expert reviewer and improved the paper as recommended.
1. Introduction
The gender and performance differences in relation to race distance of ultra-cycling, which are the purpose of this study, may have already clarified in Reference 2, 9 and 10.
Answer: The purpose of this study is to examine gender gap mainly in dependence of increasing age and racing distance in ultra-cycling. There exist papers about gender difference in general (Reference 2, 9 and 10) in ultra-cycling but there doesn`t exist a paper that answers the dynamic of gender difference in dependence of age and racing distance.
Reviwer
If so, please describe exactly what has been resolved and what remained unclear in the prevous study on the effects of race distance and increasing age on male and female in ultra cycling.
2.2. Data sampling
As the basic information, the number and average age of male and female subjects in each age group and race distance should be described.
Answer: We agree with the expert reviewer and amended the number of male and female participants. Unluckily it is not possible to mention average age because most of the races make just age-group available. The number of participating athletes has been added. (line 264-266):
The number of participating athletes by calendar year for men and women showed a highest participation of men in all distance races. Being 11,347 (91.04 %) men, and 1,116 (8.95 %) women (Figure 1).
Reviwer
OK
It is written that data from 1983 to 2018 was used, but Fig.1 shows only data from 1996 to 2018.
Answer: We agree with the expert reviewer and changed it, the data of the year 1983 had to be excluded because of missing age of the participants (corrections in line 13, 179, 384, 532).
Reviewer
・line 384 and 532 are not found
When only participants were male, it was unclear that how was data processed.
Answer: In year without female data, men data was handled in separate models (one level-no gender interaction).
Reviewer
Since the main purpose of your study is the comparison men and performance, Should the data be excluded if participants are male only?
2.3. Statistical analysis
The authors should test the gender differences in different age groups and race distances by a two-way analysis of variance?
Answer: The second two-way ANOVA model (gender x race distance) show significant effect for gender (F= 36.1; p< 0.001), race distance (F= 98.1; p<0.001), and interaction (F=13.7; p< 0.001). Line 255.
Reviwer
・The required post hoc test has not been performed when ANOVA showes a significant F value
・Line 255??
It should be noted that the performance ratio was calculated as average race speed in men/average race speed in women.
Answer: We agree with the expert reviewer and added this information for better understanding (line 18-19, 258).
Reviewer
・Line 258?
Results
Why are the age intervals different from each other? The age intervals should have been divided into 18-29, 30-39, 40-49, 50-59, and over 60 years old from standpoint of comparison with previous research results.
Answer: The recommended age group is not consistent with the age-groups of the ultra-cycling association (https://ultracycling.com/)
Reviewer
I understand
In some years, women run faster than men (Figure 2). The description that " Men were faster than women in all distance throughout the years" may be incorrect.
Answer: We agree with the expert reviewer and we changed this formulation to “in the majority” (line 387).
Reviewer
・I understand
・line 387??
The authors should test main effects and interactions in changes of average race speed of men and women with increase of race distance (Figure.3).
Answer: We agree with the expert reviewer, additional information has been inserted (line 317-320).
The second model (gender´race distance) showed a gender effect (F = 34.2, p < 0.001, h2p < 0.01), race distance effect (F = 747.7, p < 0.001, h2p = 0.15) and interaction effect (F = 16.5, p < 0.001, h2p < 0.01); pairwise comparisons showed that men were faster than women in 100-miles and 200-miles races (Figure 3).
Reviewer
You must perform post hoc test
In Figure 4, age-related change of average speed should be shown for men and women, with increase of age should be shown
Answer: Has been adapted, line 33-339:
Reviewr
・Line 33-339?
Figure 4. Performance trend by gender and age-groups of participants in ultra-distance cycling from 100- to 500-mile race. ***: significant effect (p < 0.05) for gender, age-group and interaction; *#: significant effect (p < 0.05) for age-group and interaction; *: significant effect (p < 0.05) for age-group; **: significant effect (p < 0.05) for gender and age-group.
Reviewer
Which age group showed a significant gender difference as a result of the post hoc test?
In Figure 5, the comparison of average cycling speeds between different age groups should be performed from standpoints of race distance and gender. In addition, the change of performance ratio in lower panel is different from above-mentioned definition.
Answer: Figure 5 has been excluded.
Reviewer
I understand
The description "Men showed a higher average cycling speed than women in race distances (100- to 500-mile race), with a decreasing performance ratio in the 200-mile (Figure 5)" cannot be derived from Figure 5.
Answer: We agree with the expert reviewer it cannot be derived from Figure 5 but from Figure 2. Therefore, this formulation has been changed to (line 317-319): Pairwise comparisons showed that men have a higher average cycling speed than women in all age-groups, and that men reach a higher speed at 35-44 before dropping, whereas women reach a peak at 18-34 (Figure 5).
Reviewer
・but from Figure 2 ??
・(line 317-319) is not found
Average speeds in Fig. 4 and Fig.5 should be shown by gender and race distance separately, and analyzed for main effects and interactions.
Answer: We agree with the expert reviewer and changed as requested.
Reviewer
Already commented
Discussion
4.1. Women reduce the gender gap in longer ultra-cycling in race distance
・There is no discussion on the result that the gender differences in performance observed in this study were smaller than those of previous studies. Therefore, the validity of results does not remain unclear.
Answer: We agree with the expert reviewer and changed as requested (line 3944-396).
A first important finding was that women were able to reduce the gender gap in longer distance limited ultra-cycling races (14.04 % in 100-mile races, 2.78 % in 200-mile races, 2.13 % in 400-mile races, 3.88 % in 500-mile races).
Reviewer
It is essential to compare the results with those of previous studies and examine the validity of the results of this study.
4.3. Average cycling speed and performance ration in dependence of gender and age
・The authors should clarify how authors handled male data in the calendar year when there was no female participation.
Answer: Thank you for this input, we added this information in “Statistical Analysis”. In year without female data, men data was handled in separate models (one level-no gender interaction).
Reviewer
Already commented
・According to the definition of performance ratio(men/women), the description " Comparing gender, performance ratio dropped down for the age category 34-44 years, followed by a significant increase from the age of 45 years" is not correct.
Answer: Performance ratio is only possible to compare with other age-groups. It is the ratio between averagespeedmen/averagespeedwomen. Performance ratio is a single data for each age-group.
Reviewer
According to the difinition of performance ratio : average speed men/average speed women、the description 「Focusing on gender gap, ・・increased for age-category・・60(PR:0.9178」may not be possible.
・It should be discussed whether the performance ratio between men and women have changed significantly only at the age of 34-45 years.
Answer: We have been interested how the performance ratio changes in dependence of the racing distance. Performance ratio is an additional analysis to help to interpret the results from ANOVA. Performance ratios are absolute numbers from the data they don`t have dispersion measure. This means it does not have a statistical test to compare.
Reviewer
Please delete above comment because it is based on the previous Figure of 5.
4.4. Limitations, strength and implications for future research
・The authors should show that the limitations presented do not significantly influence the conclusion in your study.
Answer: We agree with the expert reviewer and adapted this, line 529.
Nevertheless the finding in this study isn`t affected by those missing information.
Reviewer
・line 529??
・You have to explain why you think so.
Conclusions
・It may be inappropriate statement that "In distance limited ultra-cycling races held from 1983 to 2018", because the data from 1983 to 1995 was not showed.
Answer: We agree with the expert reviewer and corrected it to 1996-2018. The other data (1983-1995) had to be excluded. (corrections in line 13, 179, 384, 532).
Reviewer
・I understand corrections
・line 384, 532 ??
The hypothesis of this study was followed as the gender gap will narrow the longer the race distance. However, there is almost no change in the performance ratio from 200 to 500 miles. Therefore, the conclusion that "Female ultra-cyclist seem to be able to narrow the gap to men in longer ultra-marathon cycling distance" should be reexamined
Answer: We agree with the expert reviewer and reexamined the conclusion, gender gap will narrow in longer race distances, line 534-538.
In distance limited ultra-cycling races held from 1996 to 2018, men were faster than women for all race distances. Gender gap was lower in 400- and 500-mile races compared to the shorter races. Gender gap gets smaller in distance limited ultra-cycling races in dependence of race distance and age group.
Reviewer
・Gender gap was lower from 400- and 500-mile race→.Why 200 mile-race are not included. I don’t know why.
・If age group is the independent variable, the gender gap in each race distances
does not seem obvious as shown in Figure 4.
Author Response
Reviewer 2
Please reexamine the description in the introduction, methods, results, discussion and conclusions.
Answer: We agree with the expert reviewer and adapted those.
Abstract: This study examined a large dataset of ultra-cycling race results to investigate the sex difference in ultra-cycling performance (100 to 500 miles) in dependence of age and race distance. Data from the time period 1996-2018 were obtained from the online available database of the ultra-cycling marathon association (UMCA) including distance-limited ultra-cycling races (100-, 200-, 400- and 500-miles). A total of 12,716 race results were analyzed to compare performance between men and women by calendar year, age group (18-34, 35-44, 45-59 and 60+ years), and race distance. Men were faster than women in 100- and 200-mile races. No sex differences were identified for the 400- and 500-mile races. The performance ratio (average speedmen/ average speedwomen) dropped from 100- to 200-mile race and remained stable in 400- and 500-mile. In all race distances, the difference in average cycling speed between women and men decreased with increasing age. Women were able to close the gender gap to men in several distance-limited ultra-cycling races such as 400- and 500-mile races.
Reviewer
You need to write the conclusions of your study that answer your research purposes. The conclusions are different from the results, such as 400- and 500-mile races → Why do you exclude 200-mile race?
Answer: We agree with the expert reviewer and adapted the whole conclusion, (see line 204-208). In distance limited ultra-cycling races (100-500 miles) held from 1996 to 2018, men were faster than women for all race distances. Men showed a higher participation (91.04 %) compared to women (8.95 %). Gender gap was the highest in 100-miles races and narrowed for longer race-distances (200-, 400- and 500 miles). Gender gap gets smaller in distance limited ultra-cycling races in dependence of rising race distance and age group.
The description In all race distances, the difference in average cycling speed between women and men decreased with increasing age should be reconsidered.
Answer: We agree with the expert reviewer and adapted this statement. Concerning average cycling speed by age group, it drops from one age group to another in 100- and 200-miles races, whereupon there is no similar trend in the 400- and 500-mile races to find.
This article has the following many serious problems and was judged to be at inappropriate level as an original paper.
Answer: We agree with the expert reviewer and improved the paper as recommended.
1. Introduction
The gender and performance differences in relation to race distance of ultra-cycling, which are the purpose of this study, may have already clarified in Reference 2, 9 and 10.
Answer: The purpose of this study is to examine gender gap mainly in dependence of increasing age and racing distance in ultra-cycling. There exist papers about gender difference in general (Reference 2, 9 and 10) in ultra-cycling but there doesn`t exist a paper that answers the dynamic of gender difference in dependence of age and racing distance.
Reviewer
If so, please describe exactly what has been resolved and what remained unclear in the previous study on the effects of race distance and increasing age on male and female in ultra-cycling.
Answer: We agree with the expert reviewer and adapted the purpose of this study for better understanding (see line 72-76). For short: such a high number of athletes over such a long period of time and including that much races (45 ultra-cycling races) hasn`t been examined so far. Therefore, this study gives relevant and substantial information about gender gap in ultra-cycling.
Zingg, M.; Knechtle, B.; Rüst, C.A.; Rosemann, T.; Lepers, R. Age and gender difference in non-drafting ultra-endurance cycling performance - the 'Swiss Cycling Marathon'. Extreme Physiology and Medicine 2013, 2, doi:10.1186/2046-7648-2-18.
Based on existing literature we know that endurance performance of women in general declines to a greater event compared to men [8]. The present study relies on an enormous number of surveyed ultra-cycling races to analyze trends in ultra-cycling with special regard to gender gap in dependence of race distance and age. Until now there doesn`t exist a study including that much participants distributed to different race distances (100 miles - 500 miles).
2.2. Data sampling
As the basic information, the number and average age of male and female subjects in each age group and race distance should be described.
Answer: We agree with the expert reviewer and amended the number of male and female participants. Unluckily it is not possible to mention average age because most of the races make just age-group available. The number of participating athletes has been added. (line 264-266):
The number of participating athletes by calendar year for men and women showed a highest participation of men in all distance races. Being 11,347 (91.04 %) men, and 1,116 (8.95 %) women (Figure 1).
Reviewer
OK
It is written that data from 1983 to 2018 was used, but Fig.1 shows only data from 1996 to 2018.
Answer: We agree with the expert reviewer and changed it, the data of the year 1983 had to be excluded because of missing age of the participants (corrections in line 13, 179, 384, 532).
Reviewer
・line 384 and 532 are not found
Answer: We agree with the expert reviewer: This must be line 13, 85, 148, 179, 196 and 204.
When only participants were male, it was unclear that how was data processed.
Answer: In year without female data, men data was handled in separate models (one level-no gender interaction).
Reviewer
Since the main purpose of your study is the comparison men and performance, Should the data be excluded if participants are male only?
Answer: Dear expert reviewer, this is a good point. For specific gender comparisons, the software (SPSS) automatically ignore data if invalid for one the factors (gender). Since "gender" is factor in all three models, this was not an issue for the outcome.
2.3. Statistical analysis
The authors should test the gender differences in different age groups and race distances by a two-way analysis of variance?
Answer: The second two-way ANOVA model (gender x race distance) show significant effect for gender (F= 36.1; p< 0.001), race distance (F= 98.1; p<0.001), and interaction (F=13.7; p< 0.001). Line 255.
Reviewer
The required post hoc test has not been performed when ANOVA shows a significant F value
Answer: We agree with the expert reviewer, post-hoc analysis was now performed for all models with significant F values and added in the results. Please see all changes marked as red.
It should be noted that the performance ratio was calculated as average race speed in men/average race speed in women.
Answer: We agree with the expert reviewer and added this information for better understanding (line 18-19, 258).
Reviewer
・Line 258?
Answer: We agree with the expert reviewer, it is line 18, 102, 173.
Results
Why are the age intervals different from each other? The age intervals should have been divided into 18-29, 30-39, 40-49, 50-59, and over 60 years old from standpoint of comparison with previous research results.
Answer: The recommended age group is not consistent with the age-groups of the ultra-cycling association (https://ultracycling.com/)
Reviewer
I understand
In some years, women run faster than men (Figure 2). The description that " Men were faster than women in all distance throughout the years" may be incorrect.
Answer: We agree with the expert reviewer and we changed this formulation to “in the majority” (line 387).
Reviewer
・I understand
Answer: We agree with the expert reviewer, it is line 151 (there was a problem with formation, therefore the false lines).
The authors should test main effects and interactions in changes of average race speed of men and women with increase of race distance (Figure.3).
Answer: We agree with the expert reviewer, additional information has been inserted (line 317-320).
The second model (gender´race distance) showed a gender effect (F = 34.2, p < 0.001, h2p < 0.01), race distance effect (F = 747.7, p < 0.001, h2p = 0.15) and interaction effect (F = 16.5, p < 0.001, h2p < 0.01); pairwise comparisons showed that men were faster than women in 100-miles and 200-miles races (Figure 3).
Reviewer
You must perform post hoc test
Answer: Please see previous comment.
In Figure 4, age-related change of average speed should be shown for men and women, with increase of age should be shown
Answer: Has been adapted, line 33-339:
Reviewer
Answer: We agree with the expert reviewer, it is line 141 (Figure 4).
Figure 4. Performance trend by gender and age-groups of participants in ultra-distance cycling from 100- to 500-mile race. ***: significant effect (p < 0.05) for gender, age-group and interaction; *#: significant effect (p < 0.05) for age-group and interaction; *: significant effect (p < 0.05) for age-group; **: significant effect (p < 0.05) for gender and age-group.
Reviewer
Which age group showed a significant gender difference as a result of the post hoc test?
Answer: We agree with the expert reviewer that this information is crucial. Thus, we added this in figures and results as following: "Moreover, post-hoc analyses showed that men are faster in age-groups 18-34, 35-44 and 60+ in 100 miles races; in 200 miles races men are faster only in age-group 45-59, in 400 and 500 miles races men are faster only in 18-34 age-group". Please see all changes in manuscript marked as red.
In Figure 5, the comparison of average cycling speeds between different age groups should be performed from standpoints of race distance and gender. In addition, the change of performance ratio in lower panel is different from above-mentioned definition.
Answer: Figure 5 has been excluded.
Reviewer
I understand
The description "Men showed a higher average cycling speed than women in race distances (100- to 500-mile race), with a decreasing performance ratio in the 200-mile (Figure 5)" cannot be derived from Figure 5.
Answer: We agree with the expert reviewer it cannot be derived from Figure 5 but from Figure 2 and 4. Therefore, this formulation has been changed to (line 187-188):
Concerning average cycling speed by age group, it drops from one age group to another in 100- and 200-miles races, whereupon there is no similar trend in the 400- and 500-mile races to find.
Reviewer
・but from Figure 2 ??
Answer: à Figure 2 and Figure 4
・(line 317-319) is not found
Answer: à line 187-188
Average speeds in Fig. 4 and Fig.5 should be shown by gender and race distance separately, and analyzed for main effects and interactions.
Answer: We agree with the expert reviewer and changed as requested.
Reviewer
Already commented
Discussion
4.1. Women reduce the gender gap in longer ultra-cycling in race distance
・There is no discussion on the result that the gender differences in performance observed in this study were smaller than those of previous studies. Therefore, the validity of results does not remain unclear.
Answer: We agree with the expert reviewer and changed as requested (line 3944-396).
A first important finding was that women were able to reduce the gender gap in longer distance limited ultra-cycling races (14.04 % in 100-mile races, 2.78 % in 200-mile races, 2.13 % in 400-mile races, 3.88 % in 500-mile races).
Reviewer
It is essential to compare the results with those of previous studies and examine the validity of the results of this study.
Answer: These results concerning the dynamic of gender gap cannot be found in a comparable way in other studies. Studies that investigated trends in ultra cycling are the following below (see below, also mentioned in the study), but they didn`t have the same purpose in their study.
Concerning participation and gender difference in participation we compared our results with other studies (see line 180-182).
- Zingg: Age and gender difference in non-drafting ultra-endurance cycling performance - the swiss cycling marathon.
- Shoak MA: Participation and performance trends in ultra-cycling.
- Rüst CA: Gender difference in cycling speed and age of winning performers in ultra-cycling -the 508-mile Furnace Creek from 1983 to 2012.
4.3. Average cycling speed and performance ration in dependence of gender and age
・The authors should clarify how authors handled male data in the calendar year when there was no female participation.
Answer: Thank you for this input, we added this information in “Statistical Analysis”. In year without female data, men data was handled in separate models (one level-no gender interaction).
Reviewer
Already commented
・According to the definition of performance ratio(men/women), the description " Comparing gender, performance ratio dropped down for the age category 34-44 years, followed by a significant increase from the age of 45 years" is not correct.
Answer: Performance ratio is only possible to compare with other age-groups. It is the ratio between averagespeedmen/averagespeedwomen. Performance ratio is a single data for each age-group.
Reviewer
According to the definition of performance ratio: average speed men/average speed women、the description 「Focusing on gender gap, ・・increased for age-category・・60(PR:0.9178」may not be possible.
Answer: We agree with the expert reviewer and changed this sentence (see below, see line 190-193). Focusing on gender gap in dependence of age, the performance ratio is the lowest for the age category 34-44 years (PR: 0.8163), increases for the age-category 45-59 years (PR: 0.8818) and even more for the age-category 60+ (PR: 0.9178) showing the narrowing of gender gap in ultra-cycling with increasing age.
Reviewer
Please delete above comment because it is based on the previous Figure of 5.
We agree with the expert reviwer and deleted as recommended.
4.4. Limitations, strength and implications for future research
・The authors should show that the limitations presented do not significantly influence the conclusion in your study.
Answer: We agree with the expert reviewer and adapted this, line 529.
Nevertheless, the finding in this study isn`t affected by that missing information.
Reviewer
・line 529??
Answer: à line 197
・You have to explain why you think so.
Answer: Nevertheless, the finding in this study isn`t affected by that missing information, because we focused on measurable variables as “cycling speed, race time, race distance”. In the end it is not realistic to take account of every variable that has an impact in sports performance, we had to focus on the most important ones that can be measured and compared by each other. Additionally, it isn`t realistic to take much more other factors into consideration when you analyze 12`000 athletes. And on the other hand, when we compare gender gap - factors such as weather will influence on both gender at the same way and therefore will have the same effect on both men and women and no impact on gender gap in the end.
Conclusions
・It may be inappropriate statement that "In distance limited ultra-cycling races held from 1983 to 2018", because the data from 1983 to 1995 was not showed.
Answer: We agree with the expert reviewer and corrected it to 1996-2018. The other data (1983-1995) had to be excluded. (corrections in line 13, 179, 384, 532).
Reviewer
・I understand corrections
・line 384, 532 ??
Answer: We agree with the expert reviewer: This must be line 13, 85, 148, 179, 196 and 204.
The hypothesis of this study was followed as the gender gap will narrow the longer the race distance. However, there is almost no change in the performance ratio from 200 to 500 miles. Therefore, the conclusion that "Female ultra-cyclist seem to be able to narrow the gap to men in longer ultra-marathon cycling distance" should be reexamined
Answer: We agree with the expert reviewer and reexamined the conclusion, gender gap will narrow in longer race distances, line 534-538.
In distance limited ultra-cycling races held from 1996 to 2018, men were faster than women for all race distances. Gender gap was lower in 400- and 500-mile races compared to the shorter races. Gender gap gets smaller in distance limited ultra-cycling races in dependence of race distance and age group.
Reviewer
・Gender gap was lower from 400- and 500-mile race→.Why 200 mile-race are not included. I don’t know why.
Answer: We agree with the expert reviewer and have corrected this statement (see line 206).
Gender gap was the highest in 100-miles races and narrowed for longer race-distances (200-, 400- and 500 miles). Gender gap gets smaller in distance limited ultra-cycling races in dependence of rising race distance and age group.
・If age group is the independent variable, the gender gap in each race distances
does not seem obvious as shown in Figure 4.
Answer: We agree with the expert reviewer. In the Figure 4 you see obviously, that gender gap is the most expanded in 100-mile races and gets narrower for 200, 400- and 500-mile races.